# Estimating pulse wave velocity from the radial pressure wave using machine learning algorithms

**Weiwei Jin**[1]*, **Philip Chowienczyk**[2], **Jordi Alastruey**[1,3]

**1** Department of Biomedical Engineering, King's College London, London, United Kingdom, **2** Department of Clinical Pharmacology, St. Thomas' Hospital, King's College London, London, United Kingdom, **3** World-Class Research Centre, Digital Biodesign and Personalized Healthcare, Sechenov University, Moscow, Russia

* weiwei.jin@kcl.ac.uk, weiweijin722@gmail.com

**Data Availability Statement:** Most of the measurement data from the clinical study population, the Twins UK cohort is available for external researchers via an application (https://twinsuk.ac.uk). As the authors of this paper have not participated in the Twins UK study, we do not

## Abstract

One of the European gold standard measurement of vascular ageing, a risk factor for cardio-vascular disease, is the carotid-femoral pulse wave velocity (cfPWV), which requires an experienced operator to measure pulse waves at two sites. In this work, two machine learning pipelines were proposed to estimate cfPWV from the peripheral pulse wave measured at a single site, the radial pressure wave measured by applanation tonometry. The study populations were the Twins UK cohort containing 3,082 subjects aged from 18 to 110 years, and a database containing 4,374 virtual subjects aged from 25 to 75 years. The first pipeline uses Gaussian process regression to estimate cfPWV from features extracted from the radial pressure wave using pulse wave analysis. The mean difference and upper and lower limits of agreement (LOA) of the estimation on the 924 hold-out test subjects from the Twins UK cohort were 0.2 m/s, and 3.75 m/s & -3.34 m/s, respectively. The second pipeline uses a recurrent neural network (RNN) to estimate cfPWV from the entire radial pressure wave. The mean difference and upper and lower LOA of the estimation on the 924 hold-out test subjects from the Twins UK cohort were 0.05 m/s, and 3.21 m/s & -3.11m/s, respectively. The percentage error of the RNN estimates on the virtual subjects increased by less than 2% when adding 20% of random noise to the pressure waveform. These results show the possibility of assessing the vascular ageing using a single peripheral pulse wave (*e.g.* the radial pressure wave), instead of cfPWV. The proposed code for the machine learning pipelines is available from the following online depository (https://github.com/WeiweiJin/Estimate-Cardiovascular-Risk-from-Pulse-Wave-Signal).

## Introduction

Vascular ageing is a result of the age-induced damage inflicted upon the vascular structure and function, which leads to increased risk of chronic diseases, such as cardiovascular disease (CVD), and type 2 diabetes [1, 2]. Reducing the risk factors related to vascular ageing (*e.g.*

have the rights to share their data. The database of virtual subjects can be found in the following depository: https://github.com/peterhcharlton/pwdb/wiki/Using-the-Pulse-Wave-Database. The scripts for the machine learning pipelines proposed in this study are also available on the following online depository: https://github.com/WeiweiJin/Estimate-Cardiovascular-Risk-from-Pulse-Wave-Signal.

**Funding:** This work was supported by the British Heart Foundation (BHF) [PG/15/104/31913], the EPSRC [EP/K031546/1], the Wellcome/Engineering Physical Sciences Research Council (EPSRC) Centre for Medical Engineering at King's College London [WT 203148/Z/16/Z], the Department of Health through the National Institute for Health Research (NIHR) Cardiovascular MedTech Co-operative at Guy's and St Thomas' NHS Foundation Trust (GSTT), the comprehensive Biomedical Research Centre and Clinical Research Facilities awards to Guy's and St Thomas' NHS Foundation Trust in partnership with King's College London and King's College Hospital NHS Foundation Trust, and the Ministry of Science and Higher Education of the Russian Federation within the framework of state support for the creation and development of World-Class Research Centers "Digital biodesign and personalized healthcare" [075-15-2020-926]. The views expressed are those of the authors and not necessarily those of the BHF, Wellcome Trust, EPSRC, NIHR, GSTT or Ministry of Science. WJ was funded by a King's College London PGR International Scholarship.

**Competing interests:** The authors have declared that no competing interests exist.

blood pressure, glycemia, and lipids) at an early stage could prevent further progression of the disease [3]. Further studies have also shown that vascular ageing is associated with lifestyles [4] and exercise [5]. Thus, detecting vascular ageing at an early stage can lead to early intervention and prevention of the relevant diseases.

Studies have shown that arterial stiffening as a result of lacking compliance function, is one of the main players in vascular ageing [6, 7]. It has been suggested that arterial stiffness can be evaluated through the measurement of pulse wave velocity (PWV) [8, 9], for which the European standard assessment is the carotid-femoral PWV (cfPWV) [10]. Despite its wide use, cfPWV requires measurements at two arterial sites, manually handling the probes, and estimating the distance between the carotid and femoral arteries, which makes the measurement operator dependent. A single-site and automated measurement could overcome the limitations of the current clinical assessment of arterial stiffening.

Machine learning methods have been applied to solve a range of medical challenges, including detecting CVD. The majority of the machine learning research involving medical signals is based on either electrocardiogram (ECG) [11, 12] or photoplethysmogram (PPG) [13] data. Those studies mainly focused on critical CVD that could lead to mortality, such as heart failure [14, 15]. Whereas, the development of CVD is a long process, and early detection and intervention can stop disease progression and avoid expensive medical cost and mortality [16]. Using machine learning methods to detect earlier signs of CVD would be beneficial in improving cardiovascular health. Although little effort has been carried out to assess the CVD risk via machine learning methods, researchers have recently become engaged in the subject. For instance, a recent study has proposed a potential algorithm to estimate the size of an abdominal aortic aneurysm from pressure waves measured at carotid, brachial and femoral arteries using deep learning models [17]. In vascular ageing research, Tavallali *et al.* used an artificial neural network to estimate cfPWV with an RMSE of 1.1244 m/s. However, their approach required a central pressure wave, the carotid pressure wave, and also included other medical record information, such as chronological age [18].

This study aims to estimate cfPWV (hereafter referred to as PWV) from only the pulse wave measured at a single peripheral site (*i.e.* the radial artery) using machine learning algorithms. The following three case studies are considered. **Case Study 1** proposes a machine learning pipeline that uses Gaussian process regression to estimate PWV from key features (timing and magnitude of the fiducial points and the heart rate) extracted from the radial pressure wave measured in the Twins UK cohort. **Case Study 2** presents a second machine learning pipeline that uses a recurrent neural network (RNN) with long short-term memory (LSTM) to estimate PWV from the entire radial pressure waveform, also on the Twins UK cohort. **Case Study 3** assesses the ability of the RNN model to estimate PWV from the radial pressure waveform from a database of virtual subjects, with random noise added. Both machine learning pipelines presented in this article are available from the following online depository (https://github.com/WeiweiJin/Estimate-Cardiovascular-Risk-from-Pulse-Wave-Signal).

## Case Study 1: PWV estimation from radial pressure wave features
### Methods

**Study population.**   The study population in Case Study 1 consisted of 3,082 unselected twins (99% are females) from the Twins UK cohort. The mean and standard deviation of the biological characteristics of these subjects can be found in Table 1. The study was approved by the St Thomas' Hospital Research Ethics Committees, and all subjects signed the written informed consent. Most of the measurement data from the Twins UK cohort are available for

**Table 1. Biological characteristics of the subjects from the Twins UK cohort (N = 3,082).**

|  | Mean ± SD |
| --- | --- |
| Height (cm) | 163.2 ± 22.5 |
| Weight (kg) | 69.2 ± 27.1 |
| BMI (kg/m$^2$) | 26.2 ± 18.1 |
| Age (year) | 57.8 ± 12.8 |
| DBP (mmHg) | 74.1 ± 8.9 |
| SBP (mmHg) | 126.5 ± 17.5 |
| MAP (mmHg) | 93.6 ± 11.9 |
| PWV (m/s) | 9.39 ± 2.18 |

SD: Standard Deviation; BMI: body mass index; DBP: diastolic blood pressure; SBP: systolic blood pressure; MAP: mean arterial pressure; PWV: pulse wave velocity.

external researchers via an application. More information about this cohort can be found on its official website (https://twinsuk.ac.uk) and relevant publications [19, 20]. The data used in this case study were the radial pressure waves measured by applanation tonometry and cfPWV measured by SphygmoCor CvMS. The data were acquired by an experienced operator over the period 2006 to 2017.

**Wave feature extraction.** The features of the radial pressure wave were extracted as the timings and magnitudes of the fiducial points identified on the waveform and the heart rate using the pulse wave analyser developed by Charlton *et al.* [21]. In total, 14 fiducial points on each waveform were identified, which made the numbers of the features from one radial pressure wave to be 29. More detailed descriptions of the fiducial points can be found in the previous studies by Charlton *et al.* [21, 22].

**Preprocessing for Gaussian process regression.** Before performing the Guassian process regression, LASSO regression was performed to identify the key features from all extracted features of the waveform. Then principal component analysis (PCA) was performed after LASSO regression to exclude outliners in the analysed dataset, as the outliers could affect the accuracy of machine learning algorithms [23]. The linear model module from the scikit-learn package was used to perform the LASSO regression in Python. The hyperparameter in the model was found by 5 fold cross-validation using the GridSearchCV library. Then, PCA was performed on the key features that were identified by the LASSO regression using the PCA library from the scikit-learn package. Finally, based on the distance of the data points away from the origin, outliers were identified and excluded from the machine learning training and testing.

**Gaussian process regression.** Gaussian process regression was used to estimate PWV based on the key features from the radial pressure wave identified by LASSO regression. The advantages of using Gaussian process regression are i) it can provide uncertainty of the estimation, which most machine learning regression methods are not able to; and ii) the hyperparameters in the model can be identified by maximising the log likelihood, which is less time consuming than cross-validation. The GaussianProcessRegressor library and kernel functions from the scikit-learn package were used to perform Gaussian process regression in Python. Three kernel functions: radial basis function (RBF), Matérn kernel with $v = 5/2$, rational quadratic kernel, and their sum combinations were tested (results shown in S1 Fig). Finally, the rational quadratic kernel was chosen for this study based on the accuracy of its estimation.

**Other machine learning methods.** To confirm the accuracy of the PWV estimation by Gaussian process regression, three other machine learning methods were also used to estimate the PWV: support vector regression (SVR), and two tree-based methods (*i.e.* random forest

regression and gradient boosting regression). All machine learning algorithms were performed using the libraries from the scikit-learn package. The hyperparameters in the SVR were tuned by 5 fold cross-validation with 10 iterations using the optunity package. The hyperparameters in the tree-based methods were tuned by 10 fold cross-validation with 100 iterations using random search from the scikit-learn package. In addition, apart from the tree-based methods, the features from the radial pressure wave were normalised using the StandardScaler library in the scikit-learn package. The training and testing/developing data ratio for all machine learning analyses was 7:3.

**Error evaluation.** The root mean square error (RMSE) was calculated to evaluate each machine learning approach, which is defined as,

$$\text{RMSE} = \sqrt{\frac{\sum_{i=1}^{n} \left(\hat{\text{PWV}}_i - \text{PWV}_i\right)^2}{n}}, \tag{1}$$

where $n$ is the size of the test dataset; $\hat{\text{PWV}}_i$ and $\text{PWV}_i$ are the $i$th estimated and measured PWV, respectively. Then, a percentage error, $\epsilon$, was calculated based on the RMSE:

$$\epsilon = \frac{\text{RMSE}}{\overline{\text{PWV}}} \times 100\%, \tag{2}$$

where $\overline{\text{PWV}}$ is the mean value of the PWV of the study population.

## Results

The features from the radial pressure wave were reduced from 29 to 17 after performing the LASSO regression. The fiducial points containing those key features are shown in Fig 1a. Then, PCA was performed on the subjects using only those key features (Fig 1b). The results show that 3 of the 3,082 subjects were outliers.

The Gaussian process regression was performed on the study population without the outliers (3,079 data samples). The model was trained on 2,155 data samples. The estimation results and errors when testing on the hold-out test data set containing 924 samples are shown in Fig 2a and 2c, and Table 2, respectively. Fig 2a shows a linear relationship between the estimated and measured PWV, with a slope of 1.00 and an offset of 0.24 m/s. The coefficient of determination, $r^2$ equals to 0.42, and the p-value is less than 0.0001. The Bland-Altman plot shows a mean difference of 0.2 m/s, and upper and lower limits of agreement (LOA) of 3.75 m/s & -3.34 m/s, respectively. (Fig 2c). Both plots suggest that the accuracy of the PWV estimates deteriorated as the value of PWV increased. Table 2 illustrates that PWV could be estimated from the radial pulse waveform with an RMSE of 1.82 m/s and a percentage error, $\epsilon$, of 19.4% over the whole test data set. In addition, Gaussian process regression can also provide a statistically meaningful range (95% confidence interval) that shows the reliability of the estimation (S2 Fig).

To confirm the accuracy of the estimation made by Gaussian process regression, three other machine learning methods were applied to the same training and hold-out testing data set to estimate PWV. Table 2 shows the error evaluations of all these methods. The other three machine learning methods can provide a PWV estimation with smaller errors than Gaussian process regression, with gradient boosting regression achieving the lowest RMSE (1.63 m/s) and $\epsilon$ (17.4%). However, the reduction of the errors was limited (less than 0.2 m/s for RMSE, and less than 2% for $\epsilon$). Moreover, these alternative methods can not provide reliability of the PWV estimation (*i.e.* 95% confidence interval), and take longer to train ($\leq$ 1 minute vs $\geq$ 30 minutes). In addition, the measured PWV ploted against estimated PWV and Bland-Altman plots simulated using these three algorithms can be found in S3 Fig.

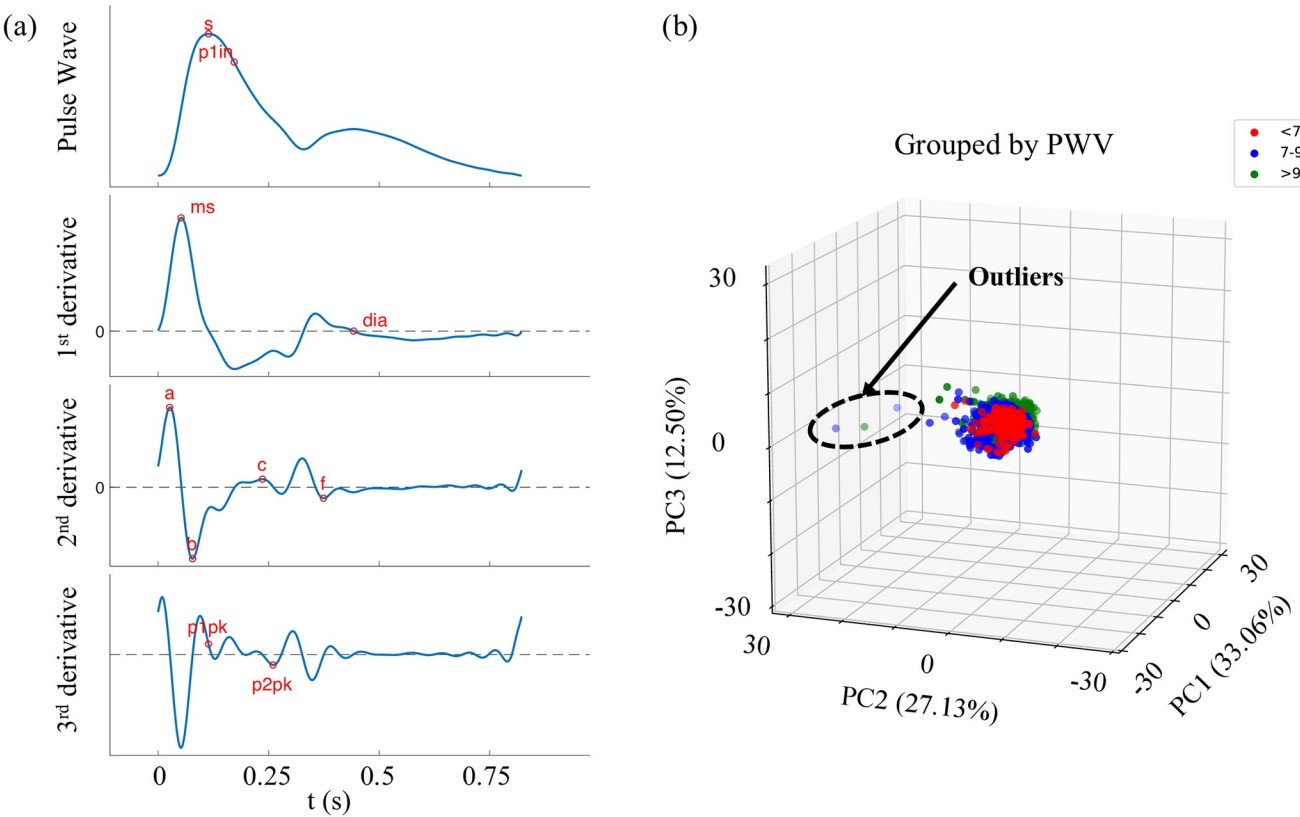

**Fig 1. Data pre-processing for pulse wave velocity estimation from the features extracted from the radial pressure wave.** (a) The fiducial points containing key features identified by the LASSO regression. (b) Identified outliers in the database using principal component analysis (PCA). Red, blue and green dots represent subject groups with pulse wave velocity (PWV) less than 7 m/s, 7–9 m/s, and greater than 9 m/s, respectively.

The Pearson's correlation coefficient, r, was used to investigate if the accuracy of the estimations using Gaussian process regression could be related to the biological characteristics. The following biological characteristics were studied: height, weight, body mass index (BMI), age (chronological age), diastolic blood pressure (DBP), systolic blood pressure (SBP), and mean arterial pressure (MAP). Fig 2e shows that the difference (between the estimated and measured PWV) correlates with the age the most, r = 0.286.

## Case Study 2: PWV estimation from the entire radial pressure wave

### Methods

The study population in Case Study 2 is identical to the study population in Case Study 1. The same error evaluation metrics were used to assess the accuracy of PWV estimation. This case study used a RNN model which is described next.

**Recurrent neural network.** The schematics of the RNN structure used in this Case Study is shown in Fig 3. The input data was an array of pressure values describing the radial pressure waveform at discrete time points. As the cardiac cycle of different subjects varied, the time duration of the radial pressure wave also differed from subject to subject. To overcome the length difference in the input data, the waves with shorter durations were extended to the duration of the longest wave by filling the array with dummy values (maximum floating point number in this case) at the end. Then, a masking layer was applied to exclude the dummy

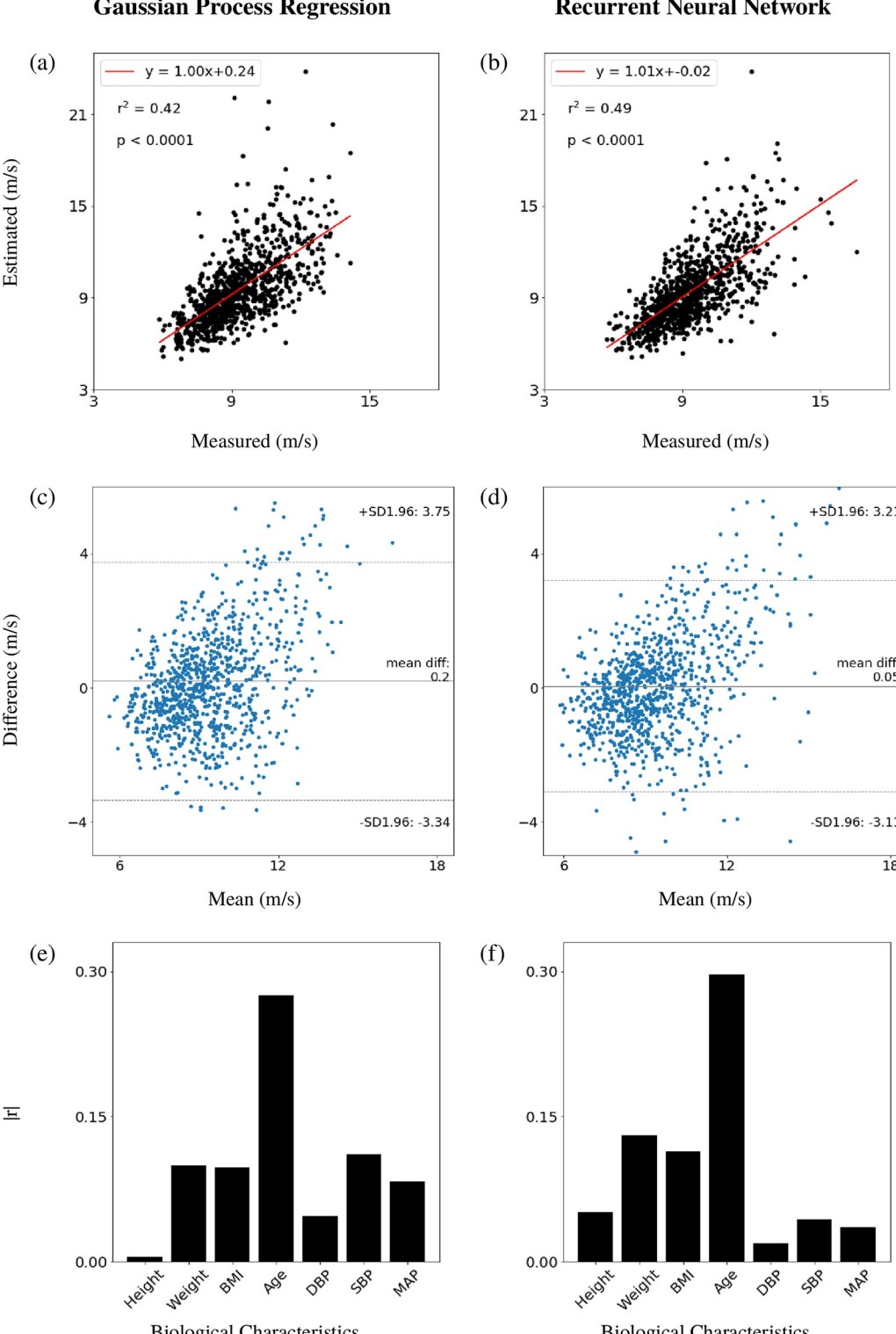

**Fig 2. Estimation of pulse wave velocity (PWV) on a hold-out test set containing 924 subjects using Gaussian process regression and recurrent neural network with long short-term memory.** Panels (a) and (b) show estimated PWV against measured PWV with the linear regression line in red, the coefficient of determination, $r^2$, and the p-value. Panels (c) and (d) show the Bland-Altman plots comparing the estimated and measured PWV. Panels (e) and (f) show Pearson correlation coefficients (r) between the biological characteristics of the cohort and the "Difference" values shown on panels (c) and (d), respectively. BMI: body mass index; DBP: diastolic blood pressure; SBP: systolic blood pressure; MAP: mean arterial pressure.

**Table 2. Root mean square error (RMSE) and percentage error ($\epsilon$) on the estimated pulse wave velocity (PWV) using different machine learning methods.**

| | RMSE (m/s) | $\epsilon$ (%) |
|---|---|---|
| Gaussian Process Regression | 1.82 | 19.4 |
| Support Vector Regression | 1.74 | 18.5 |
| Random Forest Regression | 1.64 | 17.4 |
| Gradient Boosting Regression | 1.63 | 17.4 |
| RNN | 1.59 | 16.9 |

values from being considered when estimating PWV. Afterwards, a bidirectional RNN with LSTM was used to process the time-variant radial pressure waveform, as it has been proven effective in forecasting time series data [24–26]. Finally, a dense layer with a linear activation function was used to estimate PWV based on the results from the bidirectional RNN with LSTM. Before carrying out the main simulation, hyperparameter tuning was undertaken and the following parameters were chosen: number of units for LSTM = 16; batch size = 64; epoch number = 1,500; optimizer = Adam. The RNN was constructed using open-source neural-network library TensorFlow Core v.2.2.0, including a high-level application programming interface Keras. The training and testing/developing data ratio for the RNN was also 7:3.

## Results

The RNNs with LSTM were trained and tested on the same datasets as the one used in Case Study 1. Fig 2b and 2d show the performance of the RNN. In comparison with the PWV estimation using Gaussian process regression, the RNN led to a smaller offset on the regression line (0.02 m/s vs 0.24 m/s) and a stronger correlation ($r^2$: 0.49 vs 0.42). The Bland-Altman plots show that both mean difference and the upper and lower LOA are smaller than the corresponding values obtained by Gaussian process regression (Fig 2c and 2d) (0.05 m/s vs 0.2 m/s; 3.21 m/s & -3.11 m/s vs 3.75 m/s & -3.34 m/s). The RMSE and percentage error, $\epsilon$, of PWV

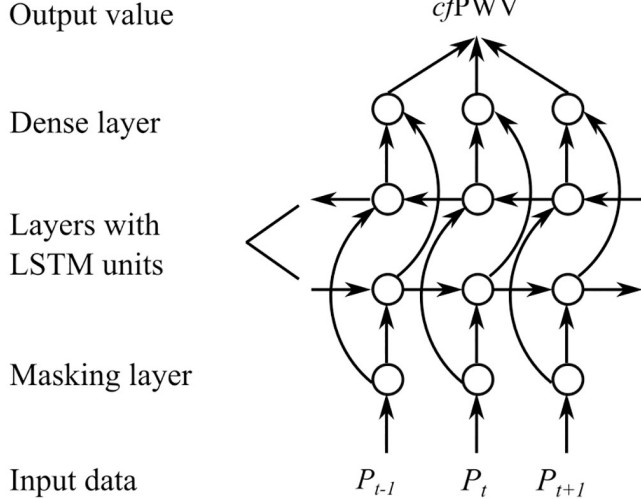

**Fig 3. Schematic illustration of the recurrent neural network structure used to estimate pulse wave velocity from the entire radial pressure wave.** $P_{t-1}$, $P_t$ and $P_{t+1}$ are the radial pressure values at the discrete time points $t-1$, $t$, and $t+1$, $cf$PWV is the carotid-femoral pulse wave velocity.

estimates using the RNN were similar to those obtained by the other machine learning methods used in Case Study 1 (see Table 2). Furthermore, Pearson's correlation coefficients, r, between biological characteristics and the difference of measured and estimated PWV calculated for the RNN model (Fig 2f) were similar to the ones obtained using Gaussian process regression, with the age again showing the strongest correlation, r = 0.297.

## Case Study 3: PWV estimation from the entire radial pressure wave with added random noise

### Methods

This case study used the RNN model described in Case Study 2, with the same training and testing/developing data ratio. The same error evaluation metrics as in Case Studies 1 and 2 were used. The following two subsections describe the student population and random noise generation.

**Study population.** To systematically investigate the effects of high-frequency noise on the radial pressure wave, a database containing 4,374 virtual subjects representative of a sample of "healthy" adults aged between 25 and 75 years old in ten-year increments was used as the study population. The database can be downloaded from the following depository: https://github.com/peterhcharlton/pwdb/wiki/Using-the-Pulse-Wave-Database. The data used in this case study were the radial pressure waves and cfPWV. Further details of this database can be found in a previous study [22]. The rational behind choosing a database of virtual subjects was to eliminate the possible effects of measurement errors.

**Noise generation.** Different intensities of high-frequency Gaussian white noise were generated and added to the radial pressure waves to test the noise sensitivity of the PWV estimation by RNN. The intensity of the noise was defined using the signal to noise ratio (SNR), similar to the approach in [27], for which the SNR was calculated as,

$$\text{SNR} = \frac{P_{\text{signal}}}{P_{\text{noise}}}, \tag{3}$$

where $P_{\text{signal}}$ and $P_{\text{noise}}$ are the power (averaged amplitude) of the pressure signal and Gaussian white noise, respectively. Six different SNRs were considered: 20, 16, 12, 10, 8 and 5. Fig 4 shows the effect of SNRs of 20, 10 and 5 on the original pressure signal.

### Results

The radial pressure waves from the database of virtual subjects augmented with different levels of random Gaussian white noise were used to test the noise sensitivity of the PWV estimation produced by the RNN model. The measured PWV plot against estimated PWV and Bland-Altman plots of the estimations from the original radial pressure wave and with SNRs of 20, 10 and 5 are shown in Fig 5. The coefficient of determination, $r^2$ for all cases considered were $\geq 0.98$. The mean difference did not increase, but the upper and lower LOA increased from 0.14 m/s & -0.24 m/s to 0.5 m/s & -0.56 m/s when adding 20% noise to the original radial pressure wave (SNR = 5). The RMSE increased from 0.10 m/s to 0.24 m/s, and the percentage error, $\epsilon$, increased from 1.2% to 2.8%, when adding 20% noise to the original radial pressure wave (Table 3). Besides, the errors of the PWV estimates using waveforms without added noise from the database of virtual subjects were over 10 times smaller than those obtained from the Twins UK cohort using the same RNN model (Table 2).

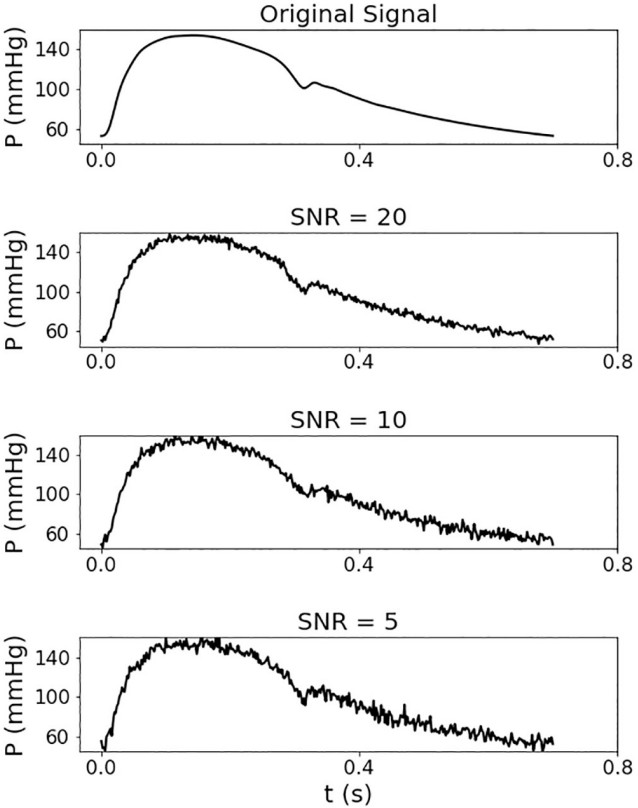

**Fig 4. An example of an original signal, and the same signal with added white noise, with signal to noise ratios (SNR) of 20, 10 and 5.**

## Discussion

We have shown the feasibility of estimating PWV from the radial pressure wave using (i) Gaussian process regression applied to features extracted from the waveform and (ii) a RNN model applied to the entire waveform. The results show that the PWV can be estimated from both pipelines, with the second pipeline presenting a slightly higher accuracy and a lower bias in the estimated PWV. However, the improvement in accuracy for PWV estimation from the second pipeline was limited, which indicated that the features extracted from the radial pressure wave using the pulse wave analyser developed by Charlton et al. [22] may be sufficient to describe the morphology of the entire radial pressure wave. Some of the key features identified by LASSO regression and applied to the PWV estimation using Gaussian process regression have been used to calculate pulse wave indices that are closely related to vascular ageing [28–30]. For instance, the reflection index can be calculated from the feature 'dia'; the augmentation index and augmentation pressure can be calculated from the features 'p1in' and 'p2pk'; and the modified ageing index is related to the features 'a', 'b', and 'c' calculated from the second derivative of the waveform. Besides, Gaussian process regression can provide a statistically meaningful range (95% confidence interval) that shows the reliability of the estimation, and required less time to train (less than a minute using the data from the Twins UK cohort). On the other hand, in order to use the pulse wave analyser to extract features from the wave, the wave needs to be preprocessed to eliminate high and low frequency noises. This step, which can result in losses of information is not required by the proposed RNN model, even when using noisy pressure waves.

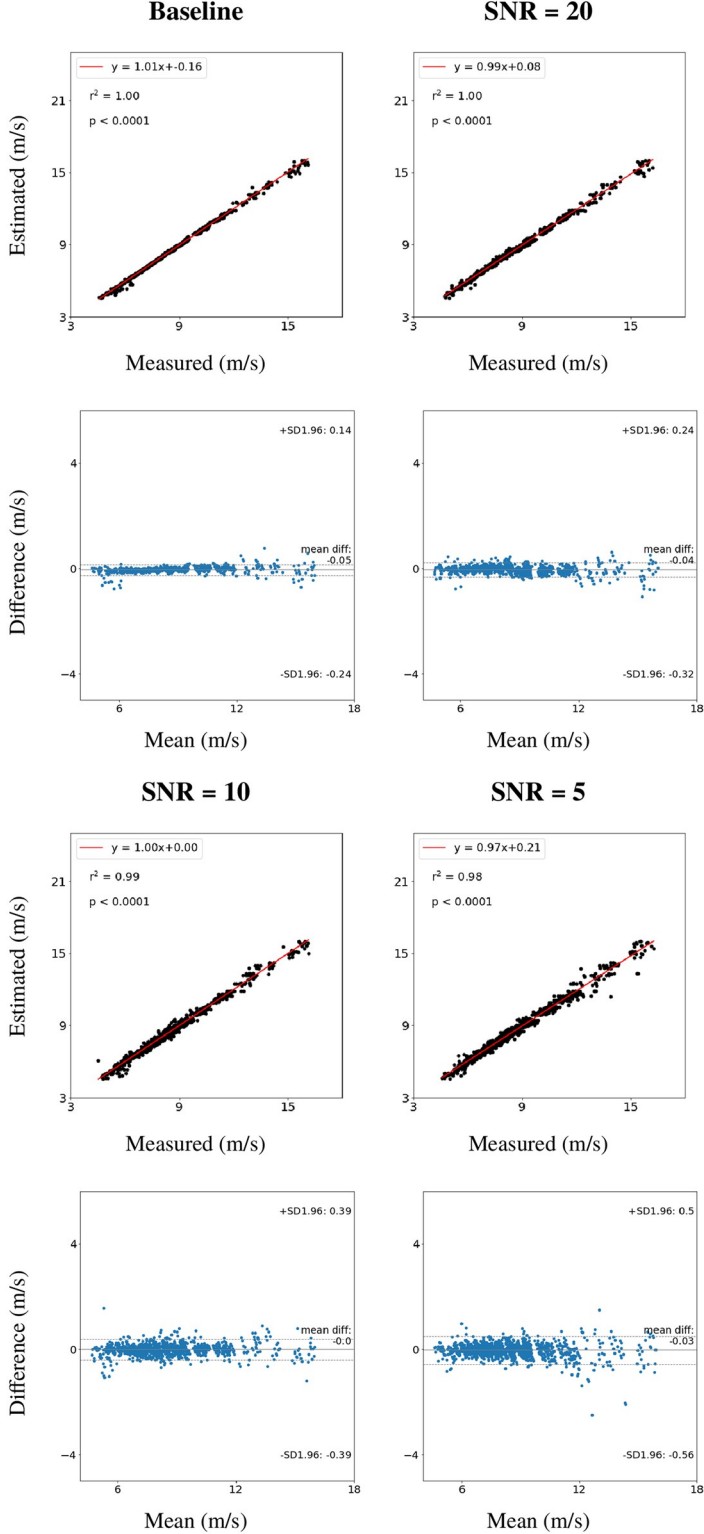

**Fig 5. Estimation of PWV on a hold-out test set containing 1312 virtual subjects using the recurrent neural network, with different levels of added white noise.** Estimated against measured PWV with the linear regression line in red, the coefficient of determination, $r^2$, and the p-value (top). Corresponding Bland-Altman plots (bottom). SNR: signal to noise ratio.

**Table 3. Root mean square error (RMSE) and percentage error ($\epsilon$) for the pulse wave velocity (PWV) estimation from the radial pressure wave by the recurrent neural network (RNN), with different intensities of added white noise.**

|  | RMSE (m/s) | $\epsilon$ (%) |
|---|---|---|
| Baseline | 0.10 | 1.2 |
| SNR = 20 | 0.15 | 1.8 |
| SNR = 16 | 0.16 | 1.9 |
| SNR = 12 | 0.16 | 1.9 |
| SNR = 10 | 0.20 | 2.4 |
| SNR = 8 | 0.21 | 2.5 |
| SNR = 5 | 0.24 | 2.8 |

Comparing our resluts with those obtained by using other non-invasive devices (*e.g.* the Pulse Pen [31]) and measurement methods (*e.g.* the oscillometric method [32]) that require pulse wave measurements in two arterial measurement sites, the mean differences between the estimated and measured PWV were similar or smaller ($\leq$ 0.214 m/s for Pulse Pen, 0.4 m/s for oscillometric method, vs $\leq$ 0.2 m/s in this study). The upper and lower LOA, however, were larger in this study ($\leq$ 1.346 m/s & $\geq$ -0.918 m/s for Pulse Pen, $\leq$2.9 m/s & $\geq$ -2.0 m/s for oscillometric method, vs $\leq$ 3.75 m/s & $\geq$-3.34 m/s in this study). When comparing our results to those obtained by using a non-invasive device that only requires a single pulse wave measurement (*e.g.* the Arteriograph [33]), the mean difference was the same for the estimation using Gaussian process regression (= 0.2 m/s), and the upper and lower LOA were smaller in this study ($\leq$ 4.5 m/s & $\geq$ -4.01 m/s vs $\leq$ 3.75 m/s & $\geq$ -3.34 m/s). The root mean square error (RMSE) of our PWV estimation was larger than that obtained in the machine learning study by Tavallali *et al.* [18] (RMSE = 1.1244 m/s). This may be explained by the fact that the average PWV in Tavallali *et al.*'s study was smaller than in this study, and that less patient information (*e.g.* chronological age) and neither the information from central arteries (*e.g.* carotid artery) were used in this study.

Based on the ARTERY Society guidelines for validation of non-invasive haemodynamic measurement devices [34], the mean differences obtained by the proposed algorithms are both "excellent", whereas the "poor" standard deviations are due to the lack of data for subjects with high PWV in the Twins UK cohort. We now discuss possible causes that led to the PWV estimate errors in our study. Firstly, the reference PWV measurements may have been inaccurate. Previous studies [35, 36] have pointed out that the accuracy of the PWV measurement can be largely affected by inaccuracies in the distance between the carotid and femoral arteries, which is measured on the patients' body surface by tape when using the SphygmoCor CvMS device. A further study showed that the accuracy of PWV measured by the SphygmoCor device decreased at higher PWV values. A possible explanation could be the larger variability of measured pulse wave transit time compared with other methods [37]. Higher PWV values are associated with small transit times, making the PWV values more sensitive to the variability in the transit time (which appears in the denominator of the PWV calculation). The RMSE and percentage error for the PWV estimates by the RNA model applied to the database of virtual subjects with noise-free data were considerably smaller (0.10 m/s vs 1.59 m/s and 1.2% vs 16.9%). This suggests the existence of measurement errors in the reference PWV values from the Twins UK cohort. However, further investigations on the accuracy of the PWV measurement would be needed to test this hypothesis. Secondly, the errors of the PWV estimates increased with the increasing PWV values, which could be due to the low number of high PWV samples in the dataset. It is known that the accuracy of machine learning algorithms decreases with the

decreasing sample size [38]. Two experiments were carried out to confirm this. First, we increased the training dataset in Case Study 1 with high PWV values by resampling the original training dataset with replacement (S4a Fig). As shown in S4b–S4f Fig, this experiment reduced the bias in the estimation for high PWV values to some extent. However, the estimation accuracy (upper and lower LOA) did not improve, since no new information was added to the training process. In the second experiment, we reshuffled the whole dataset from Case Study 1 and split the training and testing datasets with an increased number of subjects with high PWV in the training dataset. This modification improved the estimation accuracy, which brought the standard deviation produced by the RNN model to the "acceptable" level according to the ARTERY Society guidelines [34] (S5 Fig). Therefore, both the bias and the accuracy of the estimation could be improved by training the algorithms with a training database containing more subjects with high PWV values. Lastly, the errors in the PWV estimation could also be the result of confounding biological characteristics of the patients, as the radial pressure wave was the only input used in our estimation pipelines. The Pearson's correlation coefficients, r, between those biological characteristics and the difference of the estimated and measured PWV indicated that the chronological age was associated with the estimation error the most. However, this was expected since PWV has a positive correlation with chronological age and, as pointed out previously, the PWV estimation accuracy worsened for subjects with higher PWV values due to low sample numbers in the training datasets. Nevertheless, Pearson's correlation coefficients in both machine learning approaches were smaller than 0.3, indicating a negligible linear correlation [39]. Thus, the analysis suggested that the errors in the estimations would not be largely dependent on the biological characteristics.

This study is also subject to a few limitations and requires further work. Firstly, the majority of participants in the Twins UK cohort are females, which means the trained model in this study is less likely to fit well when using unseen data from a wider population. However, this should not affect the accuracy of the estimation within the analysis performed in this study and the conclusions. Secondly, the peripheral pulse wave used in this study was the radial pressure wave measured by applanation tonometry. Further studies using peripheral pulse waves, such as the PPG signal measured at the digital artery using a fingertip probe or smart phone camera, or the PPG signal measured around the wrist using the Apple Watch or Fitbit would be needed to further test the pipelines proposed in this study. Lastly, the pulse wave data in this study only contained a single cardiac cycle. Further investigations will be needed to assess the effectiveness of the RNN model on estimating cardiovascular indices using a pulse wave containing multiple cardiac cycles. The SyphygmoCor and a wearable devices such as the Apple Watch can acquire pulse wave signals over multiple cardiac cycles.

The clinical significance of this study aligns with assessing the risk factors for CVD from more accessible measurements. Firstly, the only input information to the proposed algorithms is the radial pressure wave, which is a peripheral pulse wave that can be easily measured via non-invasive devices. Importantly, this also makes the PWV estimation in this study totally independent of chronological age, which has been taken as input in other studies [18]. As chronological age does not necessarily correspond to the biological age [40], adding age as a predictor to the algorithm could also bias the estimation results. Estimating PWV without including chronological age also makes the prediction from the proposed algorithms in this study more robust and adequate for assessing vascular ageing. Secondly, the machine learning pipelines proposed in this study can also take other peripheral pulse waves, such as PPG signals, even the single lead ECG signals with more than one cardiac cycle as input to estimate CVD risks. Thirdly, the machine learning pipelines proposed in this study can be easily extended to take multiple peripheral pulse waves as input to further improve the accuracy of estimation for CVD risks.

## Conclusion

Three case studies have been carried out to investigate the possibility of estimating PWV (a well-established biomarker) from the radial pressure wave (a peripheral pulse wave) using machine learning methods. Results have shown that PWV can be estimated either from the features extracted from the pulse wave (mean difference = 0.2 m/s, upper LOA = 3.75 m/s, lower LOA = -3.34 m/s) or the entire waveform (mean difference = 0.05 m/s, upper LOA = 3.21 m/s, lower LOA = -3.11 m/s) using a clinical database (Twins UK cohort). They also suggested that the estimation of the PWV from the entire radial pressure wave using a RNN model can still be achieved when up to 20% noise is added to the wave signal using a database of virtual subjects. However, the proposed methods need to be tested for reproducibility using independent external samples. Still, the outcome of this study can potentially help deliver vascular ageing assessment to a wider population and enable repetitive measurements that could improve the accuracy of the assessment. Further application of the machine learning pipelines proposed in this study would also help with remote patient monitoring and connected health. Additionally, the scripts for the machine learning pipelines proposed in this study are also available on the following online depository: https://github.com/WeiweiJin/Estimate-Cardiovascular-Risk-from-Pulse-Wave-Signal.

## Supporting information

**S1 Fig. Estimation of pulse wave velocity (PWV) using Gaussian process regression with different kernel functions and their sum combinations.** RBF: radial basis function; Matérn: Matérn kernel; RQ: rational quadratic kernel.
(TIF)

**S2 Fig. Estimation of pulse wave velocity (PWV) with a 95% confidence interval using Gaussian process regression on a hold-out test set containing 924 subjects.** Panel (a) shows the measured and estimated PWV plot on top of each other; panel (b) shows the first ten samples in panel (a).
(TIF)

**S3 Fig. Comparison of measured and estimated pulse wave velocity (PWV) and Bland-Altman plots using support vector regression, random forest regression and gradient boosting regression on a hold-out test set containing 924 subjects.**
(TIF)

**S4 Fig. Original training and testing data and resampled training data distribution using the Twins UK cohort data (a) and Bland-Altman plots for a hold-out test set containing 924 subjects with algorithms trained using resampled training data (b-f).**
(TIF)

**S5 Fig. Resampled training and testing data distribution using the Twins UK cohort data (a) and Bland-Altman plots for a resampled hold-out test set containing 924 subjects with algorithms trained using resampled training data (b-f).**
(TIF)

## Acknowledgments

The authors would like to thank Dr James R. Bland for discussions, especially during the methodology development.

## Author Contributions

**Conceptualization:** Weiwei Jin, Philip Chowienczyk, Jordi Alastruey.

**Data curation:** Weiwei Jin.

**Formal analysis:** Weiwei Jin.

**Funding acquisition:** Weiwei Jin, Philip Chowienczyk, Jordi Alastruey.

**Investigation:** Weiwei Jin.

**Methodology:** Weiwei Jin.

**Resources:** Philip Chowienczyk, Jordi Alastruey.

**Supervision:** Philip Chowienczyk, Jordi Alastruey.

**Validation:** Weiwei Jin.

**Visualization:** Weiwei Jin.

**Writing – original draft:** Weiwei Jin.

**Writing – review & editing:** Jordi Alastruey.

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
