## [Decision Letter · Decision Letter 0]

25 Jan 2021

PONE-D-20-40380

Estimating pulse wave velocity from the radial pressure wave using machine learning algorithms

PLOS ONE

Dear Dr. Jin,

Thank you for submitting your manuscript to PLOS ONE. After careful consideration, we feel that it has merit but does not fully meet PLOS ONE’s publication criteria as it currently stands. Therefore, we invite you to submit a revised version of the manuscript that addresses the points raised during the review process.

We look forward to receiving your revised manuscript.

Kind regards,

Alberto Milan

Academic Editor

PLOS ONE

Journal Requirements:

Reviewers' comments:

Reviewer's Responses to Questions

**Comments to the Author**

1. Is the manuscript technically sound, and do the data support the conclusions?

Reviewer #1: Yes

Reviewer #2: Yes

2. Has the statistical analysis been performed appropriately and rigorously? 

Reviewer #1: Yes

Reviewer #2: Yes

3. Have the authors made all data underlying the findings in their manuscript fully available?

Reviewer #1: Yes

Reviewer #2: Yes

4. Is the manuscript presented in an intelligible fashion and written in standard English?

Reviewer #1: Yes

Reviewer #2: Yes

5. Review Comments to the Author

Reviewer #1: In this paper, the authors proposed two machine learning methods to estimate PWV from radial pressure wave obtained with arterial tonometry, by using Gaussian process regression from features extracted from the waveform and using recurrent neural network from the entire waveform. The use of random noises on the data did not change the accuracy of results obtained by RNN analysis.

The study has the merit to clearly present methods to provide an accessible PWV estimation by peripheral waveform, with some limitations.

These are my remarks:

- The proposed methods need to be tested for reproducibility and independent sample external validation. This should be clearly stated in the conclusions. The pulse wave data analysis was performed in a single cardiac cycle, while the reference method (Sphygmocor CvMS) for measurement of cfPWV require several cardiac cycles. Could the authors perform a repeatability analysis in a subgroup of subjects considering more than one cardiac cycle?

- Figure 2 and discussion: the heteroscedasticity in the distribution is attributed by the authors to general measurement errors. I partly agree with aurthors. A reduced reproducibility and thus a possible lower accuracy of cfPWV measurement was demonstrated for tonometers such a Sphygmocor at higher PWV values (see Grillo et al. Short-term repeatability of noninvasive aortic pulse wave velocity assessment: comparison between methods and devices. American journal of hypertension, 31(1), 80-88). This is an intrinsic characteristic of measurement and due to the fact that time measurement is placed at the denominator of calculation of PWV. Were the cfPWV measurements in Twins UK cohort performed twice as currently recommended?

- Figure 2 and discussion: the distribution in Bland-Altman plots look skewed for higher values. May this cause an underestimation of PWV by algorithms for higher PWV values?

Reviewer #2: This is an interesting and well written study. The issue is of high interest for scientists and clinicians. Results could inform future approaches to develop highly efficient tools aimed at facilitating the assessment of CV risk at the population level.

Mayor concerns

1- The authors stated that: “Both plots suggested that the accuracy of the PWV estimation deteriorated as the value of PWV increased”. At a visual inspection of the Bland-Altman plot, more that an increase in dispersion at increasing PWV values (heteroscedasticity), a systematic overestimation at increasing PWV value is found, suggesting systematic bias. This could be tested by appropriate statistics (e.g. correlation analysis). Please, check and modify the results accordingly.

2- The authors wrote that “Gaussian process regression can also provide a 95% confidence interval additional to the estimated PWV, which 98% of the measured PWV values were within the 95% confidence interval range”. A similar sentence is replicated also in the discussion (“Gaussian process regression was able to provide a 95% confidence interval for each estimation that covers at least 98% of the measured PWV”). I think that these sentences should be placed in the right context because they may generate a distorted perception of very high levels of accuracy of the estimated PWV approach.

I have some concerns in considering the fact that measured PWV falls within 95% of CI range is a measure of accuracy, because accuracy is usually described in terms of absolute SD values or rather as % of explained variance. I think that LOA of 3.21 m/s and -3.11 m/s, and 49% of variance explained suggest limited accuracy. The authors could also refer to Wilkinson IB et al, Artery Research 2010;4:34-40 and rephrase the sentence (especially in the discussion) accordingly.

3- The authors found that the correlation coefficient between age and the difference of the estimated and measured PWV was high, and they suggested that adding age as a predictor could potentially improve the estimation. I have a different explaination related to my point 1. If the difference between ePWV and mPWV increases at increasing PWV, and PWV increases with age, it is quite expected that this variable (difference) has a residual co-linearity with age. Do the authors agree? Rather, it is important to emphasize the fact that the process of PWV estimation is totally independent from chronological age (differently from other approaches).

6. PLOS authors have the option to publish the peer review history of their article (what does this mean?). If published, this will include your full peer review and any attached files.

Reviewer #1: **Yes: **Andrea Grillo

Reviewer #2: No

---

## [Author Response · Author response to Decision Letter 0]

5 Mar 2021

We thank the reviewers for their encouraging and constructive comments. In this document we provide a point-by-point response to the comments.

Reviewer #1

In this paper, the authors proposed two machine learning methods to estimate PWV from radial pressure wave obtained with arterial tonometry, by using Gaussian process regression from features extracted from the waveform and using recurrent neural network from the entire waveform. The use of random noises on the data did not change the accuracy of results obtained by RNN analysis.

The study has the merit to clearly present methods to provide an accessible PWV estimation by peripheral waveform, with some limitations.

These are my remarks:

- The proposed methods need to be tested for reproducibility and independent sample external validation. This should be clearly stated in the conclusions. The pulse wave data analysis was performed in a single cardiac cycle, while the reference method (Sphygmocor CvMS) for measurement of cfPWV require several cardiac cycles. Could the authors perform a repeatability analysis in a subgroup of subjects considering more than one cardiac cycle?

Thank you for the comment. We have added the following sentence to the Conclusion section to clarify that the proposed methods would need to be tested for reproducibility using independent data samples (page 10). 

“However, the proposed methods need to be tested for reproducibility using independent external samples.”

This has also been clarified in the Discussion section with the following sentence (page 10): 

“ Lastly, the pulse wave data in this study only contained a single cardiac cycle. Further investigations will be needed to assess the effectiveness of the RNN model on estimating cardiovascular indices using a pulse wave containing multiple cardiac cycles. The SyphygmoCor and a wearable device such as the Apple Watch can acquire pulse wave signals over multiple cardiac cycles.”

Unfortunately, we do not have the raw data from the SphygmoCor CvMS containing multiple cardiac cycles to use for testing the RNN model. 

- Figure 2 and discussion: the heteroscedasticity in the distribution is attributed by the authors to general measurement errors. I partly agree with aurthors. A reduced reproducibility and thus a possible lower accuracy of cfPWV measurement was demonstrated for tonometers such a Sphygmocor at higher PWV values (see Grillo et al. Short-term repeatability of noninvasive aortic pulse wave velocity assessment: comparison between methods and devices. American journal of hypertension, 31(1), 80-88). This is an intrinsic characteristic of measurement and due to the fact that time measurement is placed at the denominator of calculation of PWV. Were the cfPWV measurements in Twins UK cohort performed twice as currently recommended?

Thank you for providing these references and explaining the possible error propagation in the PWV measurement by the SphygmoCor device. The following sentence was added to the Discussion section which contains the suggested reference (page 9): 

“A further study showed that the accuracy of PWV measured by the SphygmoCor device decreased at higher PWV values. A possible explanation could be the larger variability of measured pulse wave transit time compared with other methods [1]. Higher PWV values are associated to a small transit time, making the PWV values more sensitive to the variability in the transit time (which appears in the denominator of the PWV calculation).” 

And yes, the cfPWV measurements in Twins UK cohort was performed at least twice, as currently recommended. 

- Figure 2 and discussion: the distribution in Bland-Altman plots look skewed for higher values. May this cause an underestimation of PWV by algorithms for higher PWV values?

Thank you for pointing this out. The answer is yes; the accuracy of the PWV estimates deteriorates for higher PWV values, and this is mainly due to the number of subjects with high PWV being far less than those with lower PWV. As machine learning algorithms are data driven, the scarceness of subjects with high PWV makes estimation for higher PWV values more challenging. To better illustrate this point, two additional experiments have been carried out, which involve i) increasing the training dataset from Case Study 1 (Twins UK cohort) by resampling the original training dataset with replacement, and ii) reshuffling the whole dataset from Case Study 1 and splitting the training and testing datasets with an increased number of high PWV subjects in the training dataset. The first experiment shows that increasing the training data number (weights) for higher PWV values can reduce the bias in the estimation for the testing dataset. The second experiment shows that increasing the number of high PWV values for the training dataset – while decreasing the number of high PWV for the testing dataset – can improve the accuracy of the estimation. These results have been added to the Supplemental Information along with the following paragraph, which has been added to the Discussion (page 9).

“Two experiments were carried out to confirm this. First, we increased the training dataset in Case Study 1 with high PWV values by resampling the original training dataset with replacement (S4 Fig a). As shown in S4 Fig b-f, this experiment reduced the bias in the estimation for high PWV values to some extent. However, the estimation accuracy (upper and lower LOA) did not improve, since no new information was added to the training process. In the second experiment, we reshuffled the whole dataset from Case Study 1 and split the training and testing datasets with an increased number of subjects with high PWV in the training dataset. This modification improved the estimation accuracy, which brought the standard deviation produced by the RNN model to the ”acceptable” level according to the ARTERY Society guidelines [2] (S5 Fig). Therefore, both the bias and the accuracy of the estimation could be improved by training the algorithms with a training database containing more subjects with high PWV values.”

 

Reviewer #2

This is an interesting and well written study. The issue is of high interest for scientists and clinicians. Results could inform future approaches to develop highly efficient tools aimed at facilitating the assessment of CV risk at the population level.

Major concerns

1- The authors stated that: “Both plots suggested that the accuracy of the PWV estimation deteriorated as the value of PWV increased”. At a visual inspection of the Bland-Altman plot, more that an increase in dispersion at increasing PWV values (heteroscedasticity), a systematic overestimation at increasing PWV value is found, suggesting systematic bias. This could be tested by appropriate statistics (e.g. correlation analysis). Please, check and modify the results accordingly.

Thank you for raising this point. The reason for the underestimation for higher PWV values is due to the number of subjects with high PWV being far less than those with lower PWV. As machine learning algorithms are data driven, the scarceness of subjects with high PWV makes estimation for higher PWV values more challenging. To better illustrate this point, two additional experiments have been carried out. These involved i) increasing the training dataset from Case Study 1 (Twins UK cohort) by resampling the original training dataset with replacement, and ii) reshuffling the whole dataset from Case Study 1 and splitting the training and testing datasets with an increased number of high PWV subjects in the training dataset. The first experiment shows that increasing the training data number (weights) for higher PWV values can reduce the bias in the estimation for the testing dataset. The second experiment shows that increasing the number of high PWV values for the training dataset – while decreasing the number of high PWV for the testing dataset – can improve the accuracy of the estimation. These results have been added to the Supplemental Information along with the following paragraph, which has been added to the Discussion (page 9).

“Two experiments were carried out to confirm this. First, we increased the training dataset in Case Study 1 with high PWV values by resampling the original training dataset with replacement (S4 Fig a). As shown in S4 Fig b-f, this experiment reduced the bias in the estimation for high PWV values to some extent. However, the estimation accuracy (upper and lower LOA) did not improve, since no new information was added to the training process. In the second experiment, we reshuffled the whole dataset from Case Study 1 and split the training and testing datasets with an increased number of subjects with high PWV in the training dataset. This modification improved the estimation accuracy, which brought the standard deviation produced by the RNN model to the ”acceptable” level according to the ARTERY Society guidelines [2] (S5 Fig). Therefore, both the bias and the accuracy of the estimation could be improved by training the algorithms with a training database containing more subjects with high PWV values.”

2- The authors wrote that “Gaussian process regression can also provide a 95% confidence interval additional to the estimated PWV, which 98% of the measured PWV values were within the 95% confidence interval range”. A similar sentence is replicated also in the discussion (“Gaussian process regression was able to provide a 95% confidence interval for each estimation that covers at least 98% of the measured PWV”). I think that these sentences should be placed in the right context because they may generate a distorted perception of very high levels of accuracy of the estimated PWV approach.

I have some concerns in considering the fact that measured PWV falls within 95% of CI range is a measure of accuracy, because accuracy is usually described in terms of absolute SD values or rather as % of explained variance. I think that LOA of 3.21 m/s and -3.11 m/s, and 49% of variance explained suggest limited accuracy. The authors could also refer to Wilkinson IB et al, Artery Research 2010;4:34-40 and rephrase the sentence (especially in the discussion) accordingly.

Thank you for pointing this out. We agree that using 95% confidence interval as a metric for estimation accuracy might not be appropriate here. We have deleted this sentence from the Abstract. However, the 95% confidence interval is a statistically meaningful range that shows the reliability of the estimation. Thus, the sentence regarding the confidence interval in the Discussion (page 8) and elsewhere (page 5) has been modified to the following sentence. 

“Gaussian process regression can provide a statistically meaningful range (95% confidence interval) that shows the reliability of the estimation.”

With regards to the discussion on accuracy, the following sentence has been added to the Discussion which includes the suggested reference by Wilkinson IB et al. (page 9).

“Based on the ARTERY Society guidelines for validation of non-invasive haemodynamic measurement devices [2], the mean differences obtained by the proposed algorithms are both “excellent”, whereas the “poor” standard deviations are due to the lack of data for subjects with high PWV in the Twins UK cohort. We now discuss possible causes that led to the PWV estimate errors in our study.”

3- The authors found that the correlation coefficient between age and the difference of the estimated and measured PWV was high, and they suggested that adding age as a predictor could potentially improve the estimation. I have a different explaination related to my point 1. If the difference between ePWV and mPWV increases at increasing PWV, and PWV increases with age, it is quite expected that this variable (difference) has a residual co-linearity with age. Do the authors agree? Rather, it is important to emphasize the fact that the process of PWV estimation is totally independent from chronological age (differently from other approaches).

Thank you for your insights. Yes, we agree that PWV generally increases with age, and the difference between estimated and measured PWV increases with increasing PWV values and, thus, the chronological age. The sentence involving correlation coefficient in the Discussion has been modified as follows (page 9-10).

 “The Pearson’s correlation coefficients, r, between those biological characteristics and the difference of the estimated and measured PWV indicated that the chronological age was associated with the estimation error the most. However, this was expected since PWV has a positive correlation with chronological age and, as pointed out previously, the PWV estimation accuracy worsened for subjects with higher PWV values due to low sample numbers in the training datasets.”

And yes, the point here is that the PWV estimation in this study is totally independent from chronological age, which differs from other approaches. The following sentence has also been added to the last paragraph in the Discussion to emphasise this (page 10). 

“Importantly, this also makes the PWV estimation in this study totally independent of chronological age, which has been taken as input in other studies [3]. As chronological age does not necessarily correspond to the biological age [4], adding age as a predictor to the algorithm could also bias the estimation results. Estimating PWV without including chronological age also makes the prediction from the proposed algorithms in this study more robust and adequate for assessing vascular ageing.”

References

Repeatability of Noninvasive Aortic Pulse Wave Velocity Assessment: Comparison between Methods and Devices. Am J Hypertens. 2018;31: 80–88. doi:10.1093/ajh/hpx140

2. Wilkinson IB, McEniery CM, Schillaci G, Boutouyrie P, Segers P, Donald A, et al. ARTERY Society guidelines for validation of non-invasive haemodynamic measurement devices: Part 1, arterial pulse wave velocity. Artery Res. 2010;4: 34–40. doi:10.1016/j.artres.2010.03.001

3. Tavallali P, Razavi M, Pahlevan NM. Artificial intelligence estimation of carotid-femoral pulse wave velocity using carotid waveform. Sci Rep. 2018;8: 1–12. doi:10.1038/s41598-018-19457-0

4. Shiels PG, McGuinness D, Eriksson M, Kooman JP, Stenvinkel P. The role of epigenetics in renal ageing. Nat Rev Nephrol. 2017;13: 471–482. doi:10.1038/nrneph.2017.78

---

## [Decision Letter · Decision Letter 1]

2 Jun 2021

Estimating pulse wave velocity from the radial pressure wave using machine learning algorithms

PONE-D-20-40380R1

Dear Dr. Jin,

We’re pleased to inform you that your manuscript has been judged scientifically suitable for publication and will be formally accepted for publication once it meets all outstanding technical requirements.

Kind regards,

Alberto Milan

Academic Editor

PLOS ONE

Additional Editor Comments (optional):

Reviewers' comments:

Reviewer's Responses to Questions

**Comments to the Author**

1. If the authors have adequately addressed your comments raised in a previous round of review and you feel that this manuscript is now acceptable for publication, you may indicate that here to bypass the “Comments to the Author” section, enter your conflict of interest statement in the “Confidential to Editor” section, and submit your "Accept" recommendation.

Reviewer #1: All comments have been addressed

Reviewer #2: All comments have been addressed

2. Is the manuscript technically sound, and do the data support the conclusions?

Reviewer #1: Yes

Reviewer #2: Yes

3. Has the statistical analysis been performed appropriately and rigorously? 

Reviewer #1: Yes

Reviewer #2: Yes

4. Have the authors made all data underlying the findings in their manuscript fully available?

Reviewer #1: Yes

Reviewer #2: Yes

5. Is the manuscript presented in an intelligible fashion and written in standard English?

Reviewer #1: Yes

Reviewer #2: Yes

6. Review Comments to the Author

Reviewer #1: The paper is significantly improved from the previous version. My previous remarks have been adequately addressed.

Reviewer #2: (No Response)

7. PLOS authors have the option to publish the peer review history of their article (what does this mean?). If published, this will include your full peer review and any attached files.

Reviewer #1: No

Reviewer #2: No

---

## [Editor Report · Acceptance letter]

17 Jun 2021

PONE-D-20-40380R1 

Estimating pulse wave velocity from the radial pressure wave using machine learning algorithms 

Dear Dr. Jin:

I'm pleased to inform you that your manuscript has been deemed suitable for publication in PLOS ONE. Congratulations! Your manuscript is now with our production department. 

Kind regards, 

on behalf of

Dr. Alberto Milan 

Academic Editor

PLOS ONE